# High-Performance Graphene Nanowalls/Si Self-Powered Photodetectors with HfO_2_ as an Interfacial Layer

**DOI:** 10.3390/nano13101681

**Published:** 2023-05-19

**Authors:** Yuheng Shen, Yulin Li, Wencheng Chen, Sijie Jiang, Cheng Li, Qijin Cheng

**Affiliations:** 1School of Electronic Science and Engineering, Xiamen University, Xiamen 361102, China; 2Shenzhen Research Institute of Xiamen University, Xiamen University, Shenzhen 518000, China

**Keywords:** hafnium oxide, graphene nanowalls, plasma-enhanced chemical vapor deposition, photodetectors

## Abstract

Graphene/silicon (Si) heterojunction photodetectors are widely studied in detecting of optical signals from near-infrared to visible light. However, the performance of graphene/Si photodetectors is limited by defects created in the growth process and surface recombination at the interface. Herein, a remote plasma-enhanced chemical vapor deposition is introduced to directly grow graphene nanowalls (GNWs) at a low power of 300 W, which can effectively improve the growth rate and reduce defects. Moreover, hafnium oxide (HfO_2_) with thicknesses ranging from 1 to 5 nm grown by atomic layer deposition has been employed as an interfacial layer for the GNWs/Si heterojunction photodetector. It is shown that the high-k dielectric layer of HfO_2_ acts as an electron-blocking and hole transport layer, which minimizes the recombination and reduces the dark current. At an optimized thickness of 3 nm HfO_2_, a low dark current of 3.85 × 10^−10^, with a responsivity of 0.19 *AW*^−1^, a specific detectivity of 1.38 × 10^12^ as well as an external quantum efficiency of 47.1% at zero bias, can be obtained for the fabricated GNWs/HfO_2_/Si photodetector. This work demonstrates a universal strategy to fabricate high-performance graphene/Si photodetectors.

## 1. Introduction

Graphene is a two-dimensional material with excellent optical and electrical properties, such as extremely high carrier mobility and ultra-broadband optical absorption. The high carrier mobility enables graphene to travel for micrometers without scattering at room temperature [1]. These properties make graphene have a wide range of potential applications, including energy storage [2], polymer composites [3], biomedical science [4,5], photoelectric devices [6,7,8,9,10,11], etc. Junctions formed by graphene and silicon (Si) can function as Schottky diodes, which are widely used in solar cells and photodetectors [6,7,8,9]. The graphene/Si solar cell was first reported in 2010 with a photovoltaic conversion efficiency (PCE) of 1.5% [6], and the PCE of the graphene/Si solar cell has reached 16.2% [7] within a few years of development. Graphene has also been reported to gain significant achievements in the field of photoelectric detection, including ultrahigh photoresponsivity, ultrafast photoresponse, and ultrawideband response [8,9,10,11].

The methods of preparing graphene include mechanical exfoliation [12], chemical vapor deposition (CVD) [13], heat-induced epitaxial growth on silicon carbide (SiC) [14], etc. In 2004, Geim et al. [12] successfully exfoliated and observed a monolayer of graphene from high-directional thermal cracking graphite for the first time by mechanical exfoliation method, but mechanical exfoliation with low controllability is difficult to obtain large areas of high-quality graphene. The CVD-grown graphene, which is synthesized at a high growth temperature, high vacuum, and selective substrate, requires a catalyst that is difficult to remove, and the transfer process for the following device fabrication is fussy and complex [15]. Multilayers of graphene sheets have been prepared by catalyst-free radio-frequency (RF) plasma-enhanced chemical vapor deposition (PECVD) technique [16]. These multilayers of graphene sheets are approximately perpendicular to the surface of the substrate, forming a wall network with a corrugated surface. Therefore, this graphene network with vertical stacking can also be called vertically oriented graphene nanowalls (GNWs). GNWs with unique characteristics of ultra-sharp edges, high aspect ratio, high specific surface area, and high stability feature an excellent electrode network, which could collect photo-generated carriers quickly [17,18,19]. PECVD is a catalyst-free direct growth method, which can avoid the complex transfer process for the following device fabrication. So far, GNWs have been reported to successfully grow by PECVD on Cu [20], SiO_2_ [21], Al_2_O_3_ [22], etc. In comparison with the traditional growth methods of graphene, PECVD growth of GNWs has several advantages, such as a relatively low growth temperature, a fast growth rate, and no selectivity toward substrates, which can facilitate the application of GNWs in photodetectors and photovoltaic devices.

However, a zero bandgap and a uniform 2.3% optical absorption of the intrinsic graphene [23] result in fast recombination and a low photoresponsivity, which limits its deeper and wider application in the optoelectronics [24]. To improve the performance of graphene photodetectors, the construction of heterojunction devices through the combination of other semiconductors, such as Si, is a simple and feasible method (it is noteworthy that a lot of work has been reported on the combination of Si with other low-dimensional materials (such as WS_2_, Ag nanowalls, etc.) to form a heterojunction for the optoelectronic devices [25,26,27,28]). However, the existence of numerous defects, such as dangling bonds on the Si surface, leads to carrier recombination, which seriously affects the photoelectric performance. Modification of the contact between graphene and Si with an interfacial layer is the most popular technique for reducing the charge recombination [29]. It has been reported that the deposition of an oxide layer on the Si surface as an interfacial layer can effectively reduce the effect of surface defects [30]. Among various oxides, including SiO_2_, Al_2_O_3_, etc., hafnium oxide (HfO_2_), with a band gap of ~5.5 eV and a high dielectric constant (~25), has a relatively lower effective tunneling mass and lower valence band offset, which leads to a higher tunneling probability than SiO_2_ and Al_2_O_3_ [31].

In this work, we report a direct way to fabricate GNWs/HfO_2_/Si photodetectors. The GNWs are obtained using remote PECVD at a low-power (300 W) and catalyst-free moderate temperature (900 °C). In a typical PECVD process for the growth of graphene, both the dissociation process of the precursor gases and the growth process of graphene occur on the substrate surface at the same time. The particles with high energy often cause damage to the surface of the substrate in the process of GNWs deposition, and an unwanted ion bombardment often introduces defects and degrades the performance of the fabricated photodetectors. The low-power remote PECVD proposed in our work can be used to avoid unwanted ion bombardment. In addition to the direct deposition of GNWs without using any catalyst, both lower growth power and faster growth offer more possibilities for the large-scale production of graphene. Furthermore, to improve the photoelectric performance of the pristine GNWs/Si photodetector, we introduce a HfO_2_ interfacial layer grown by atomic layer deposition (ALD) to increase the Schottky barrier height, suppress the recombination in the interface, and passivate the dangling bonds on the Si. HfO_2_ also acts as an electron-blocking and hole transport layer in the GNWs/HfO_2_/Si photodetector, which plays an important role in reducing the dark current. To further investigate the influence of HfO_2_ on the performance of GNWs/HfO_2_/Si photodetectors, we optimize its thickness. The optimized thickness of HfO_2_ is 3 nm, and at this thickness, a low dark current of $3.85\times 10^{-10}\;A$ with a high responsivity of $0.19\;AW^{-1}$, a specific detectivity of $1.38\times 10^{12}\;Jones,$ as well as an external quantum efficiency of 47.1% under the bias of 0 V for the fabricated GNWs/HfO_2_/Si photodetector, can be obtained. Moreover, we systematically analyze the band diagram of the device and the physical mechanism of device performance improvement.

## 2. Materials and Methods

### 2.1. Growth of HfO_2_ by Atomic Layer Deposition

Since GNWs are generally p-doped in the atmosphere environment, we choose n-type Si as a starting material for the fabrication of photodetectors [15]. The n-type (100) single crystalline Si with a resistivity of 1–10 Ω·cm and a thickness of 600 μm was cleaned by RCA (Radio Corporation of America, New York, NY, USA) cleaning to remove organic contaminants and natural oxide layer and then was purged with nitrogen gas and stored under vacuum to avoid reoxidation. Before the deposition of HfO_2_, one-third area of the Si substrate was covered with heat tape to prevent the growth of HfO_2_ in this area. A series of HfO_2_ with different thicknesses were directly deposited on the Si substrate at a temperature of 150 °C using Tetrakis dimethylamido hafnium (TDMAHf) as the precursor of hafnium and H_2_O as the precursor of oxygen. In this process, N_2_ with a flow rate of 50 sccm was used as the carrier gas, and the temperature of TDMAHf was held at 75 °C while the temperature of H_2_O was at room temperature.

### 2.2. Growth of GNWs

GNWs were directly grown on the Si substrate that had already been deposited with a layer of HfO_2_ by remote PECVD. Specifically, a remote radio-frequency plasma-enhanced horizontal tube furnace deposition system has been used to grow GNWs (shown in Figure 1a). The plasma is generated at the coil position, and the frequency of the plasma generator is 13.6 MHz. It is worthwhile to mention that the plasma generator is away from the center of the tube furnace where the Si substrate was placed (the distance between the plasma generator and the center of the tube furnace is 40 cm). Before the synthesis process of GNWs, the entire tubular furnace was pumped to a pressure of approximately 10^−3^ mbar by a vacuum pump, and thereafter, the tube was heated to 900 °C. Then we introduced a mixture of methane (CH_4_) with a flow rate of 10 sccm and argon with a flow rate of 40 sccm. The plasma was turned on to grow GNWs under an RF power of 300 W (the power density was 0.093 W/cm^3^) with a growth time of 60, 90, 120, and 150 s, respectively. After the growth of GNWs, the sample was naturally cooled to room temperature.

### 2.3. Fabrication of Photodetectors

The GNWs grown for 120 s were used to fabricate GNWs/HfO_2_/Si photodetectors. GNWs were spin-coated with UV-positive photoresist (AZ 5214) and then dried at 96 °C for 4 min. Then we used a metal mask to shield the 2/3 area of the sample grown with HfO_2_. Afterward, the sample was exposed to UV light with a power of 1 W for 15 s and immersed in the developer. In this process, a photoresist, which had not been removed by the developer, was used to protect the graphene on the HfO_2_ in our experiment. After removing the excess photoresist, we purged it with oxygen plasma (60 sccm O_2_ and 80 W RF power) for 5 min to remove extra GNWs without the protection of the photoresist on Si. At this time, GNWs were only retained on the side of HfO_2_. Then, acetone was used to remove the remaining photoresist. After that, we used physical vapor deposition (PVD) to sputter metal electrodes (Cr/Au). The electrodes both kept quasi-ohmic contacts with Si and GNWs (the corresponding contact characteristic curves can be found in Appendix A). The schematic structure of the GNWs/HfO_2_/Si photodetector is shown in Figure 1b. The overall fabrication process of the GNWs/HfO_2_/Si photodetector is shown in Figure 2. In addition, the real apparatus and intermediate fabrication products are shown in Appendix A.

### 2.4. Characterization Technique

Xplora Raman Spectroscope was used to measure the Raman spectra of the GNWs with a 532 nm line of the semiconductor laser. A Zeiss Supra 55 field emission scanning electron microscope (SEM) was used to observe the surface morphology of the synthesized GNWs. The optical transmittance of GNWs was measured by UV-Vis-NIR spectrometer (UV-2600) in the wavelength range from 300 to 1400 nm. The electrical properties of the GNWs were measured by the Hall effect measurement system (HMS 5000). High-resolution transmission electron microscopy (HRTEM) images were taken on a JEOL 2100 TEM microscope operated at 300 kV. The thickness of HfO_2_ was measured using a profilometer (Bruker, DektakXT-A). The surface roughness of GNWs was characterized by S600LS atomic force microscope (AFM). The capacitance–voltage (C–V) of the fabricated Al/HfO_2_/p-Si metal–oxide–semiconductor (MOS) capacitor was measured using a Keithley 4200 semiconductor parameter analyzer. The current–voltage (I–V) characteristics of the fabricated photodetectors were measured by Keithley 4200 source meter. Illumination was generated using a light-emitting diode (LED) with a beam diameter of 4 mm and a spectral wavelength of 532 nm in air.

## 3. Results and Discussion

The surface morphology of GNWs plays an important role in the performance of the GNWs/HfO_2_/Si photodetector. Figure 3a–d shows the SEM images of GNWs grown on the Si substrate with different growth times of 60 s (a), 90 s (b), 120 s (c), and 150 s (d), respectively. As shown in Figure 3a, for the GNWs prepared at a deposition time of 60 s, the density of the GNWs on the substrate is very low, and the size of the GNWs is very small. Moreover, one can notice that the GNWs do not form a continuous film at a growth time of 60 s. Both the size and the density of GNWs have been found to increase significantly with the increase in the growth time from Figure 3a–d. Particularly, when the growth time is 150 s, we can find that all the substrate has been covered with the GNWs and that all the GNWs are interconnected. It is generally considered that with the increase in the growth time, more C_2_ radicals, which were produced from CH_4_ precursor, could have enough time to deposit on the Si substrate for the synthesis of GNWs, which made GNWs have a larger size and density [32].

Figure 3e shows the Raman spectra of GNWs grown on HfO_2_/Si with different growth times of 60, 90, 120, and 150 s, respectively. From Figure 3e, we can’t detect any Raman peak in the GNWs when their growth time was 60 s, while all other samples show the three most prominent Raman peaks, i.e., D (~1350 cm^−1^), G (~1580 cm^−1^), and 2D (~2700 cm^−1^). Additionally, the Raman peak of D’ (~1620 cm^−1^), which presents as a shoulder of the G peak in Raman spectra for GNWs prepared with 90, 120, and 150 s, can be associated with the defects and edges of graphene [33]. All samples do not have an obvious peak of D’, indicating a small amount of inter-crystalline defects [34]. The strong D peak originated from *sp*^3^ carbon clusters of the GNWs, indicating the main sources of defects in our samples [35,36,37]. The graphitized structure and the *sp*^2^ hybridization of carbon atoms in the synthesized GNWs are mainly shown in the sharp G peak of all samples [36]. Furthermore, the 2D peak represents the band structure of graphene and originates from a two-phonon double resonance process [35].

To identify the number of layers of the GNWs, we carried out HRTEM measurements. Figure 3f shows a typical HRTEM image for the GNWs grown for 120 s. One can notice that GNWs consist of multi-layered graphene. It is noteworthy that we don’t use 2D/G to determine the number of layers of the GNWs in our work. This is because 2D/G is used to calculate the number of layers in graphene materials grown in very well-controlled conditions where well-defined and homogeneous materials with very few layers (and a very narrow distribution of a number of layers) are synthesized [38]. In our case, using such a growing time in the fast-growing process used, very dense graphene material is produced, and thus, it is quite difficult to extract information from 2D/G.

Figure 3g displays the transmittance spectra of GNWs grown on clean quartz substrates with varying growth times from 60 to 150 s. As observed in Figure 3g, the transmittance of GNWs gradually decreases with an increase in the growth time. For example, the transmittance of GNWs at an incident wavelength of 550 nm, denoted as T_550_, is 96.4% for a growth time of 90 s and 91.5% for a growth time of 150 s. Table 1 presents the electrical and optical properties of GNWs grown for 90, 120, and 150 s. One can notice that the carrier concentration, mobility, and conductivity increase, while the sheet resistance decreases with an increase in the growth time. Specifically, as the growth time increases from 90 to 150 s, the sheet resistance of GNWs decreases from 254.3 to 123.5 Ω/sq., while the mobility, conductivity, and carrier concentration increase from 1.08 cm^2^V^−1^s^−1^, 0.131 Ω^−1^cm^−1^, 1.18 × 10^16^ cm^−3^ to 2.62 cm^2^V^−1^s^−1^, 0.270 Ω^−1^cm^−1^, and 6.42 × 10^17^ cm^−3^, respectively. The electrical properties of GNWs grown for 150 s are superior to those of the other two samples due to a large amount of interconnected graphene. However, the T_550_ for GNWs grown for 150 s is lower than the other two samples. The properties of GNWs grown for 120 s achieve a compromise between sheet resistance and transmittance.

Figure 4 shows the AFM images of GNWs grown on the Si substrate with different growth times of 90 s (a) and 120 s (b), respectively. From Figure 4, one can notice that the growth time can affect the surface roughness of GNWs. The root mean square (RMS) roughness of the GNWs with a growth time of 90 s is about 4.231 nm, while the RMS roughness of the GNWs with a growth time of 120 s is about 16.890 nm. The edges of GNWs are significantly larger with a longer growth time, which makes the surface rougher.

Let’s now briefly discuss the benefits of remote PECVD on the growth of GNWs. Generally, in the typical PECVD growth of GNWs, there exist two main processes, i.e., the process of gas phase reaction and the process of surface reaction. During the process of gas phase reaction, the dissociation process of the precursor gases is always accompanied by the unwanted ionization process of the precursor gases under the glow discharge. This complex reaction process would generate various free radicals (such as CH, CH_2_), ions (such as CH^+^, H^+^), and electrons [39]. Afterward, the deposition of the reactive carbon dimer C_2_ in the process of surface reaction contributes to the formation of the graphene. At the same time, active hydrogen ions (H^+^, etc.) have an etching effect on grown graphene. Remote PECVD decouples these two processes (the process of gas phase reaction and the process of surface reaction) and effectively reduces the ion bombardment, which leads to a lower defect density in GNWs [32] (optical emission spectroscopy (OES) measurement shown in Appendix A demonstrates that the ion bombardment by Ar-related radicals and the etching effect by H-related radicals are lower at the position of the surface reaction in comparison with those at the position of the plasma generation). Compared with the traditional CVD growth of graphene, there exist a large number of free radicals in the remote PECVD process, which makes it possible to grow GNWs within a few minutes.

The properties of HfO_2_ also play an important role in the GNWs/HfO_2_/Si photodetector in this work. Figure 5a,b shows height profiles of HfO_2_ grown on the Si substrate measured by profilometer for 200 and 400 cycles of ALD growth, respectively. The thickness of HfO_2_ in Figure 5a is about 21 ± 1 nm, and the thickness of HfO_2_ in Figure 5b is about 43 ± 2 nm. It is concluded that the thickness of the obtained HfO_2_ is approximately linear with the number of growth cycles. The growth rate deduced from Figure 5a,b is approximately 1.07 Å/cycle (the cross-sectional SEM image of HfO_2_ grown on the Si substrate for 500 cycles presented in Appendix A further confirms that the growth rate is approximately 1.07 Å/cycle). Figure 5c shows the AFM image of HfO_2_ grown on the Si substrate for 400 cycles. The RMS roughness of HfO_2_ films is about 0.718 nm, indicating that the HfO_2_ film is rather smooth. Figure 5d shows the UV-visible absorption spectrum of HfO_2_ grown on the clean quartz substrate for 400 cycles. For HfO_2_ with an indirect band gap, the band gap of HfO_2_ can be estimated using the following formula [40]:(1)$${\left(\alpha h\upsilon \right)}^{\frac{1}{2}}=B\left(h\upsilon -Eg\right)$$
where α is the absorption coefficient, which is derived from the absorption spectrum; hυ is the photon energy; *Eg* is the band gap, and *B* is a constant. The inset of Figure 5d shows the plot of (α*hυ*)^1/2^ versus *hυ* to extract the band gap of HfO_2_, which is roughly estimated to be 4.88 ± 0.12 eV. The C–V characteristic curve of the Al/HfO_2_/p-Si MOS capacitor is shown in Figure 5e. The dielectric constant of HfO_2_ can be obtained using the following formula [41]:(2)$$C_{ox}=A\left|\frac{dQ_{G}}{dV_{ox}}\right|=A\;\frac{\varepsilon_{0}\varepsilon_{r0}}{d_{ox}}$$
where *A* is the area of electrode (*A* is 10^−8^ m^2^ in this work); $Q_{G}$ is the total gate charge; $V_{ox}$ is the oxide voltage; $\varepsilon_{0}$ is the vacuum permittivity ($\varepsilon_{0}$ is 8.854 × 10^−12^ F/m^2^); $C_{ox}$ is the capacitance of gate oxide ($C_{ox}$ is 2.78 × 10^−10^ F obtained from Figure 5e); $d_{ox}$ is the thickness of HfO_2_ ($d_{ox}$ is 10 nm in this work), and $\varepsilon_{r0}$ is the dielectric constant of HfO_2_. Based on Equation (2), $\varepsilon_{r0}$ of HfO_2_ can be calculated as 31.398 in this work.

In this work, GNWs grown for a deposition time of 120 s were selected to fabricate the GNWs/Si photodetector due to the highest photo-to-dark current ratio (I–V curves of the fabricated GNWs/Si photodetectors for GNWs grown for 90, 120, and 150 s are shown in Appendix A). For photoelectric characterization, Figure 6a shows the I–V characteristics of the GNWs/Si Schottky junction photodetectors with or without HfO_2_ under the condition of darkness and illumination. From Figure 6a, we can find that with the introduction of HfO_2_, the dark current shows a pronounced drop from 10^−9^ to 10^−10^ A, while the photo-generated current shows an increase from 10^−8^ to 10^−7^ A under the bias of 0 V. The photo-to-dark current ratio (PDCR) of our devices with or without HfO_2_ under the bias of 0 V are 617 and 69, respectively. Clearly, our device exhibits a distinct self-powered characteristic, and this characteristic has been enhanced with the introduction of HfO_2_.

Figure 6b shows the time-dependent photoresponse of these two devices under a bias of 0 V and an incident power of 5 $\mu W/{\mathrm{cm}}^{2}$. From Figure 6b, we can find the rise time (τ_r_, which is defined as the time of the current ranging from 10–90%) and the decay time (τ_d_, which is defined as the time of the current ranging from 90–10%) both decrease slightly with the introduction of HfO_2_. Specifically, the rise time and the decay time of the GNWs/HfO_2_/Si photodetector are 0.13 and 0.14 s, respectively, while the rise time and the decay time of the GNWs/Si photodetector are 0.16 and 0.15 s, respectively. The introduction of this interface layer makes it more difficult for electrons to transport through the barrier [42]. Therefore, the recombination of holes and electrons at the interface is reduced, and the strong built-in electric field in the GNWs/HfO_2_/Si heterojunction can greatly improve the separation of the photo-generated carriers [43]. The introduction of the HfO_2_ layer here passivates the surface states of the Si, and the defects of GNWs could be reduced, resulting in the shortening of the rise time [42]. It should be noted that the time response characterization shown in Figure 6b was limited by the acquisition time resolution and that the actual response time is expected to be faster than what is shown in Figure 6b. Additionally, the carrier lifetime in GNWs is expected to be longer than in graphene, which could lead to a longer time for carriers to be collected by the electrodes [44]. Despite the promising results, the response time of the device still needs to be improved.

On the basis of previously obtained data, we can calculate the responsivity ($R_{\lambda }$) of the fabricated photodetectors using the following formula [45]:(3)$$R_{\lambda }=\frac{I_{ph}}{A*P}$$
where *I_ph_* is the photocurrent generated under light illumination, which is calculated by subtracting the current measured in the dark from the current measured under light illumination *(I*_light_–*I*_dark_); A is the active area of the device (*A* is $6.25\times 10^{-2}\;cm^{2}$ in our work), and *P* is the power density of the incident light (*P* is $5\;\mu W/{\mathrm{cm}}^{2}$ in our work). The calculated responsivity of the device without HfO_2_ is about 0.058 A/W, and the calculated responsivity of the device with HfO_2_ is about 0.19 A/W.

Specific detectivity ($D^{*}$) is also an important indicator to evaluate the performance of the photodetector, and we can get it from the following formula [25]:(4)$$D^{*}\approx \frac{\sqrt{A}R_{\lambda }}{\sqrt{2eI_{dark}}}$$
where e is the electronic charge, and *I*_dark_ refers to the dark current. The calculated values of $D^{*}$ of the fabricated photodetectors without HfO_2_ and with HfO_2_ are about 4.2 × 10^11^ and 1.38 × 10^12^ Jones at zero bias, respectively. One can notice that, after the introduction of HfO_2_ as an interfacial layer, $D^{*}$ has a significant increase.

External quantum efficiency (*EQE*) is also an important parameter to evaluate the performance of the photodetector. It can be interpreted as the ratio of the number of the photo-generated electron–hole pairs, which contributes to the photocurrent, to the total number of the incident photons. The EQE can be obtained by the following formula [45]:(5)$$EQE=\frac{Rhc}{e\lambda }$$
where *h* is the Planck’s constant (6.626 × 10^−34^ m^2^·kg/s); λ is the wavelength of the incident light (λ is 532 nm in our work); and *c* is the speed of light. The *EQE*s of the GNWs/Si and GNWs/HfO_2_/Si photodetectors are about 13.5% and 47.1%, respectively.

From the above analysis, one can notice that in comparison with the photodetector without HfO_2_, the GNWs/HfO_2_/Si photodetector has a lower dark current, a higher PDCR, and a faster response time. To explain the underlying physical mechanism behind this phenomenon, the energy band diagrams for the GNWs/Si and GNWs/HfO_2_/Si photodetectors are shown in Figure 7. Without a HfO_2_ interfacial layer, photo-generated electrons of Si can easily move toward GNWs through thermal emission because of the low built-in potential between GNWs and Si, as shown in Figure 7a. Then, the unwanted recombination occurs immediately as the short lifetime of the carriers in the GNWs [46], giving rise to the poor performance of the GNWs/Si photodetector. The existence of a HfO_2_ interfacial layer modifies the band alignment, resulting in a band bending in the valence band of Si and an increased Schottky barrier height in the interface, as shown in Figure 7b. When a thin HfO_2_ interfacial layer is introduced, the movement of the photo-generated electrons from Si toward GNWs would be significantly blocked by the increased $\varphi_{SBH}$ (the Schottky barrier height), while photo-generated holes can tunnel through HfO_2_ due to its higher probability of tunneling (after the introduction of a HfO_2_ interfacial layer, the barrier height for photogenerated electrons is significantly higher than that of photogenerated holes, leading to a higher probability of photogenerated holes than photogenerated electrons). Therefore, the recombination of the carriers in GNWs can be suppressed effectively [15,42,47,48], leading to the excellent performance of the GNWs/HfO_2_/Si photodetector. However, if the thickness of HfO_2_ is too thick, the tunneling probability of holes through HfO_2_ decreases, and therefore, holes accumulate at the interface. This will result in a higher recombination rate of photo-generated carriers and deteriorate the performance of the GNWs/HfO_2_/Si photodetector. At this time, the quasi-Fermi level for holes in the silicon shows obvious bending due to the accumulation of the holes, as shown in Figure 7c. Our experimental result demonstrates that the optimal thickness of HfO_2_ is 3 nm in our work, which will be presented later.

The increase in $\varphi_{SBH}$ can be confirmed by thermionic emission theory according to the following equation [49]:(6)$$J_{dark}V=J_{s}\left[\exp\left(\frac{eV}{nkT}\right)-1\right]$$
where *J_s_* is the reverse saturation current density; e is the elementary charge; V is the applied voltage; *n* is an ideality factor; *k* is a Boltzmann constant, and *T* is the temperature. Furthermore, *J_s_* also satisfies the following equation [47]:(7)$$J_{s}\left(T\right)=A^{*}T^{2}\exp\left(-\frac{e\left(\varphi_{SBH}\right)}{kT}\right)$$
where $A^{*}$ is Richardson constant (252 A·cm^−2^K^−2^ for n-type Si). Taking the logarithm of both sides of the Equation (6), with the assumption that $\exp\left(\frac{eV}{nkT}\right)\gg 1$, we can obtain *n* and $\varphi_{SBH}$ by fitting the linear part of the curve based on the Equations (6) and (7). Figure 8a shows the I–V characteristics of the GNWs/Si photodetectors with and without a HfO_2_ layer. Figure 8b shows the lnJ–V curves of photodetectors with and without a HfO_2_ layer. The results of the barrier height for photodetectors with and without HfO_2_ are 0.87 ± 0.02 and 0.82 ± 0.02 eV, respectively. According to a previous study [47], there is a strong positive correlation between build-in potential (V_bi_) and $\varphi_{SBH}.$ For Schottky heterojunction photodetectors, the photo-generated carriers can be separated by the built-in electric field in the depletion region [50]. A larger V_bi_ and $\varphi_{SBH}$ could facilitate the separation and migration of photo-generated carriers, which enhances the transfer of the photo-generated holes at the interface. Moreover, the increase in $\varphi_{SBH}$ blocks the thermal emission of the photo-generated electrons from the side of Si toward the side of GNWs. As a result, both the leakage current and the recombination at GNWs can be greatly reduced. The ideality factor *n* is also a critical factor that needs to be taken into consideration. The ideality factor decreases from 1.32 ± 0.02 to 1.17 ± 0.02, as shown in Figure 8b, indicating that the fabricated photodetector with HfO_2_ has a better junction quality and the recombination of the carriers has significantly reduced. Through calculation and comparison, it is found that the barrier of our devices is slightly higher than other graphene/Si-based devices [15,42] and that the value of *n* is slightly higher than an ideal Schottky junction (*n* = 1). The increase in $\varphi_{SBH}$ and the decrease in *n* further confirm the improvement of our devices due to the introduction of HfO_2_.

Last but not least, the optimization of the thickness of HfO_2_ must be taken into account for the achievement of high-performance photodetectors. Figure 9 shows the I–V curves of the GNWs/HfO_2_/Si photodetectors with different thicknesses of the HfO_2_ interface layer. Table 2 lists the calculated parameters of the GNWs/HfO_2_/Si photodetectors with different thicknesses of HfO_2_. The result of the experiment is consistent with previous theoretical analysis. The introduction of HfO_2_ increases the height of the Schottky barrier and blocks the transport of electrons toward graphene when the interface layer is thin. As shown in Figure 9a, the dark current of our devices under 0 V bias has only a slight difference with different thicknesses. Meanwhile, a thicker HfO_2_ results in a lower dark current. The current under illumination condition keeps rising under zero bias until the thickness of HfO_2_ reaches 3 nm due to the enhancement of $\varphi_{SBH}$, as shown in Figure 9b. However, there is a slight decrease when the thickness continuously increases up to 5 nm. From Table 2, the ideal factor has been reduced from 1.35 ± 0.02 to 1.17 ± 0.02, and the $\varphi_{SBH}$ increases from 0.82 ± 0.02 to 0.87 ± 0.02 eV when the thickness of HfO_2_ increases from 0 to 3 nm. However, with a further increase in the HfO_2_ interface layer to 5 nm, *n* increases to 1.33 ± 0.02, at which point the recombination of the carriers is no longer suppressed, and the $\varphi_{SBH}$ doesn’t show a continuous increase (the detailed *n* and $\varphi_{SBH}$ of the GNWs/Si photodetectors with the thickness of 1 and 5 nm HfO_2_ layer can be found in Appendix A). As a result, the PDCR of the photodetector with 5 nm thick HfO_2_ is worse than the photodetector with an ideal thickness (3 nm) of HfO_2_. Although an interface layer with thin thickness enhances the built-in potential and reduces the interface trap state density, an excessively thick oxide layer also limits the transport of the photo-generated holes. A large number of holes accumulate at the interface, resulting in the recombination at the interface and the increase in n. The variation tendencies of PDCR and responsivity are the same as n, according to Table 2. The above discussion further confirms that when the thickness of HfO_2_ is 3 nm, we can get better photodetector performance.

Performance parameters of typical graphene/Si-based photodetectors with different structures are compared in Table 3. One can notice that the specific detectivity and responsivity of our photodetector are comparable to those reported works. The reasons for the performance improvement of the GNWs/HfO_2_/Si photodetector in this work are as follows. Firstly, the remote plasma decouples the whole process of GNWs growth and independent control of precursor gas dissociation and other growth parameters. Secondly, the introduction of the HfO_2_ interface layer passivates the surface of Si and reduces the density of interface states, enhancing the junction characteristic of the GNWs/HfO_2_/Si photodetector. Finally, the precise control of the thickness of the interface layer effectively maintains a balance between electron blocking and hole transporting.

## 4. Conclusions

In this work, we successfully grew GNWs using a remote PECVD and fabricated self-powered GNWs/Si photodetectors with different structures to systematically explore the possible influencing factors of the GNWs/HfO_2_/Si photodetector. Remote plasma enhances the deposition rate and reduces structural defects of GNWs. After the introduction of the HfO_2_ interface layer with a thickness of 3 nm, the dark current decreases from 10^−9^ to 10^−10^ A, the PDCR increases from 69 to 617, and the specific detectivity increases from 4.2 × 10^11^ to 1.38 × 10^12^ Jones at the bias of 0 V. Based on the experimental result, we have also proposed a physical mechanism to shed light on the improvement of the photoelectric performance of the GNWs/Si photodetector after introducing HfO_2_ as an interface layer with an appropriate thickness. The approach of using HfO_2_ as an interfacial layer for the improvement of GNWs/Si photodetectors can be applied in other heterojunction-based photoelectric devices. Our work offers effective guidance for fabricating GNWs-based photodetectors and pushes forward the application of graphene in photodetectors.

## Figures and Tables

**Figure 1 nanomaterials-13-01681-f001:**
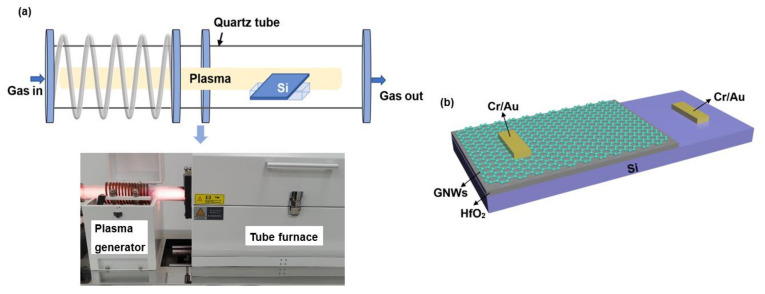
(**a**) A schematic and photo of a remote radio-frequency plasma-enhanced horizontal tube furnace deposition system, respectively. (**b**) Schematic structure of the GNWs/HfO_2_/Si photodetector.

**Figure 2 nanomaterials-13-01681-f002:**
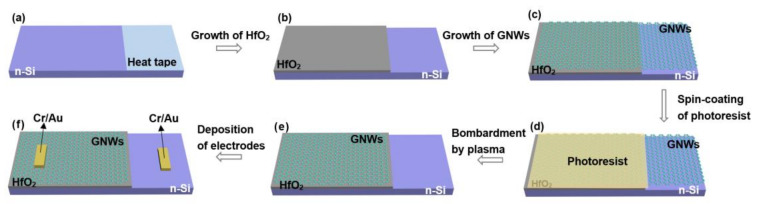
Fabrication process of the GNWs/HfO_2_/Si photodetector: (**a**) one-third of the Si substrate was shielded with heat tape; (**b**) HfO_2_ was grown by ALD on two-thirds of the Si substrate that has not been covered by heat tape; (**c**) GNWs were grown by PECVD; (**d**) photoresist was coated to protect GNWs on HfO_2_; (**e**) GNWs were removed by plasma bombardment, and photoresist was removed by acetone; (**f**) electrodes were deposited by PVD.

**Figure 3 nanomaterials-13-01681-f003:**
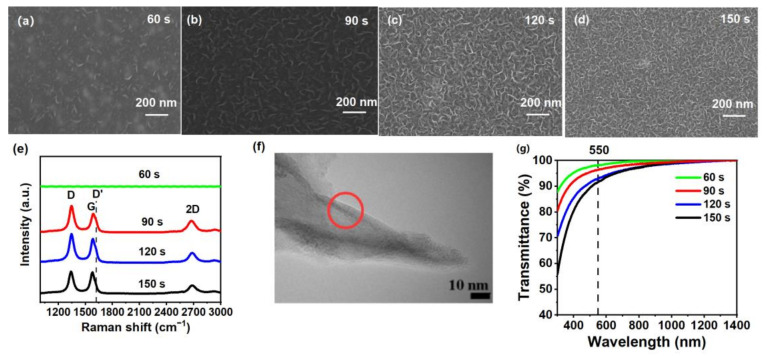
SEM images of GNWs grown on the Si substrate with different growth times: (**a**) 60 s; (**b**) 90 s; (**c**) 120 s; (**d**) 150 s; (**e**) Raman spectra of GNWs on HfO_2_/Si with different growth times (the spectra have been displaced vertically for clarity); (**f**) a typical HRTEM image for the GNWs grown for 120 s; (**g**) transmittance spectra of GNWs grown on the clean quartz substrate with different growth times.

**Figure 4 nanomaterials-13-01681-f004:**
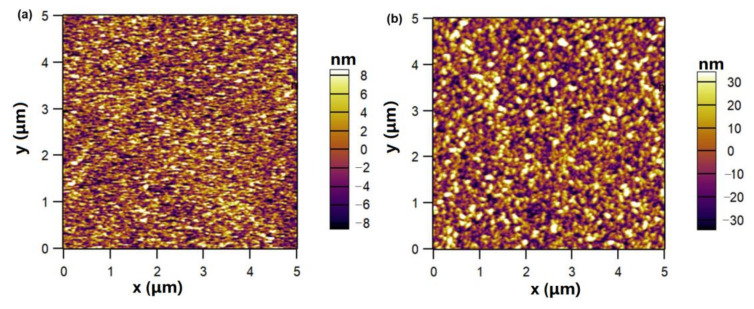
AFM images of GNWs grown on the Si substrate with different growth times: (**a**) 90 s; (**b**) 120 s.

**Figure 5 nanomaterials-13-01681-f005:**
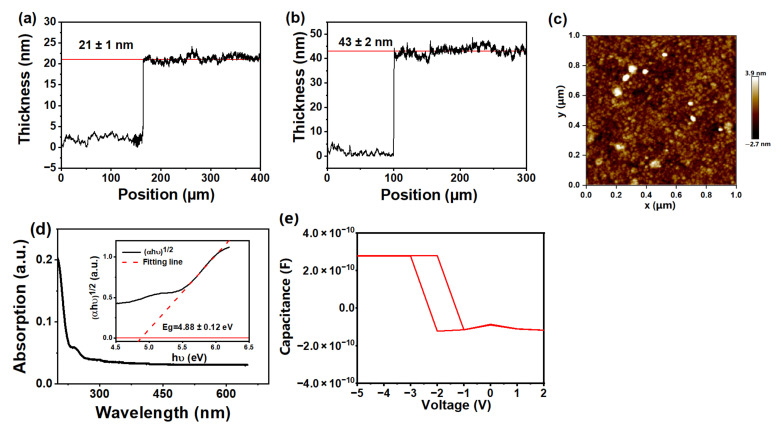
Height profiles of HfO_2_ grown on the Si substrate for (**a**) 200 cycles and (**b**) 400 cycles of ALD growth. (**c**) AFM images of HfO_2_ grown on the Si substrate for 400 cycles. (**d**) UV-visible absorption spectrum of HfO_2_ grown on the clean quartz substrate for 400 cycles, and the inset shows the plot of (αhυ)^1/2^ versus hυ (α is the absorption coefficient of HfO_2_). (**e**) C–V characteristic curve of the Al/HfO_2_/p-Si metal–oxide–semiconductor capacitor.

**Figure 6 nanomaterials-13-01681-f006:**
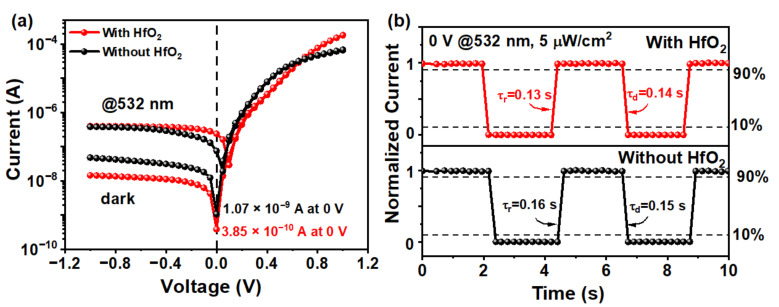
(**a**) I–V characteristics of the GNWs/Si Schottky junction photodetectors with or without HfO_2_ under the condition of darkness and illumination. (**b**) Time-dependent photoresponse of the GNWs/Si photodetectors with or without HfO_2_ under the bias of 0 V.

**Figure 7 nanomaterials-13-01681-f007:**
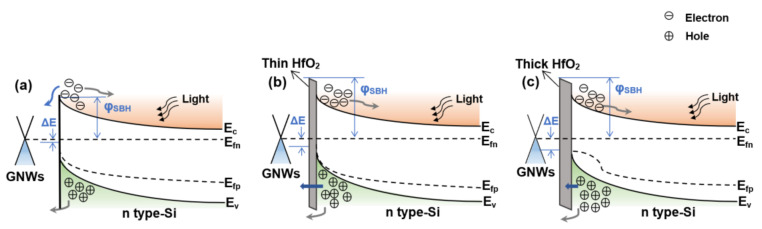
Energy band diagrams of the GNWs/Si photodetectors (**a**) without a HfO_2_ interfacial layer, (**b**) with a thin HfO_2_ interfacial layer, and (**c**) with a thick HfO_2_ interfacial layer under illumination conditions. Here $\varphi_{SBH}$ represents the Schottky barrier height; ΔE is the difference between GNWs Fermi level and quasi-Fermi energy level for holes in Si; E_c_ and E_v_ are the energy levels of conduction band and valence band for Si, respectively; E_fn_ and E_fp_ are quasi-Fermi energy levels of electrons and holes for Si, respectively.

**Figure 8 nanomaterials-13-01681-f008:**
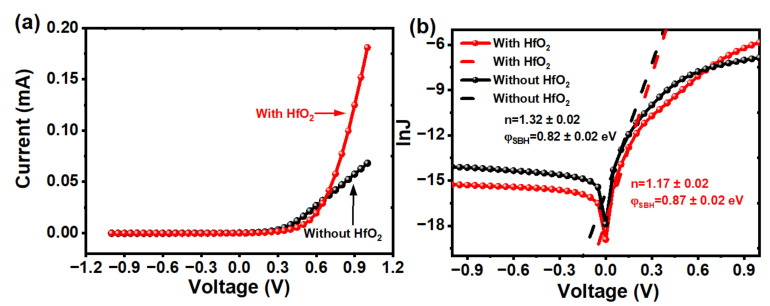
(**a**) I–V characteristics of the GNWs/Si photodetectors with and without a HfO_2_ layer. (**b**) lnJ–V curves of the GNWs/Si photodetectors with and without a HfO_2_ layer under dark conditions.

**Figure 9 nanomaterials-13-01681-f009:**
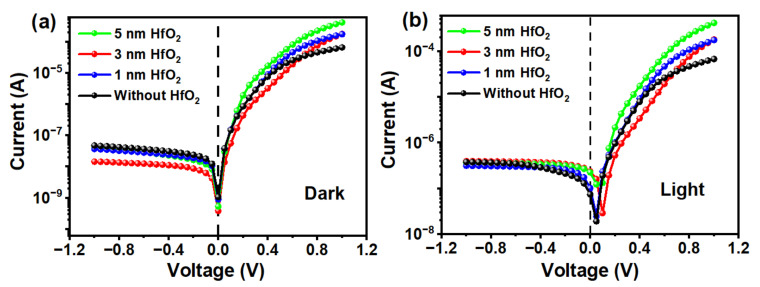
(**a**) I–V characteristic of the GNWs/HfO_2_/Si photodetectors under dark conditions and (**b**) under illumination of the devices with varying thicknesses of HfO_2_.

**Table 1 nanomaterials-13-01681-t001:** Electrical and optical properties of GNWs prepared with different growth times of 90, 120, and 150 s.

Time (s)	Sheet Resistance (Ω/sq.)	Mobility (cm^2^V^−1^s^−1^)	Conductivity (Ω^−1^cm^−1^)	Carrier Concentration (cm^−3^)	Transmittance at 550 nm
90	254.3	1.08	0.131	1.18 × 10^16^	96.4%
120	211.7	1.77	0.157	5.55 × 10^17^	93.1%
150	123.5	2.62	0.270	6.42 × 10^17^	91.5%

**Table 2 nanomaterials-13-01681-t002:** The calculated parameters of the GNWs/HfO_2_/Si photodetectors with different thicknesses of HfO_2_.

Thickness (nm)	PDCR @ 0 V	Responsivity (A/W)	Ideal Factor	Schottky Barrier Height (eV)
0	69	0.058	1.35 ± 0.02	0.82 ± 0.02
1	113	0.18	1.24 ± 0.02	0.85 ± 0.02
3	617	0.19	1.17 ± 0.02	0.87 ± 0.02
5	425	0.08	1.33 ± 0.02	0.86 ± 0.02

**Table 3 nanomaterials-13-01681-t003:** Performance comparison of different graphene/Si-based photodetectors.

Ref.	Device Structure	Responsivity (A/W)	Response Time (τ_rise_/τ_decay_)	Specific Detectivity (cm·Hz^1/2^/W)
[44]	GNWs/Si	0.012 @0 V	-	7.85 × 10^6^
[51]	GNWs/Si	0.015	43/69 μs	1.5 × 10^11^
[52]	GNWs/Si	0.52 @0 V	40 μs	5.88 × 10^13^
[53]	Graphene/Si	0.225 @−2 V	-	7.69 × 10^9^
[54]	GQDs/WSe_2_/Si	0.707 @−3 V	0.2/0.14 ms	4.51 × 10^9^
[55]	Graphene/Si	1.38×10^−4^ @0 V	0.37 ms	1.6 × 10^9^
[56]	GNWs/DLC/Si	2400	13/36 μs	1.07 × 10^11^
This work	GNWs/HfO_2_/n-Si	0.19 @0 V	0.13/0.14 s	1.38 × 10^12^

## Data Availability

The datasets used and analyzed during the current study are available from the corresponding author on reasonable request.

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
