# Peer review of "High-Performance Graphene Nanowalls/Si Self-Powered Photodetectors with HfO2 as an Interfacial Layer"

_nanomaterials, 2023, doi:10.3390/nano13101681_

Round 1
Reviewer 1 Report (Previous Reviewer 1)
The replies to my comments and the revision made to the manuscript are satisfactory
Author Response
Thank you very much for this positive comment.
Reviewer 2 Report (Previous Reviewer 4)
The authors have addressed the reviewers' comments satisfactorily. The revised manuscript can be accepted in its present form. Thank you.
Authors check the manuscript to avoid typos.
for example
"4.2×10^11 Jones to 1.38×10^12 Jones" should be "4.2×10^11 to 1.38×10^12 Jones"
"10^-9 A to 10^-10 A" should be "10^-9 to 10^-10 A".
Author Response
Thank you very much for pointing out this mistake. We have modified this mistake in Lines 17-18, Page 21 in the revised manuscript.
Reviewer 3 Report (New Reviewer)
Dear Authors
This manuscript is focused on high-performance Graphene Nanowalls/Si Self-powered photodetectors with HfO2 as an interfacial layer. A remote plasma-enhanced chemical vapor deposition is introduced to directly grow graphene nanowalls (GNWs) at a low power, which can effectively improve the growth rate and reduce defects. Moreover, hafnium oxide (HfO2) grown by atomic layer deposition has been employed as an interfacial layer for the GNWs/Si heterojunction photodetector. This article presented concerns on an interesting and actual subject. The following suggestion and comments should be taken:
1) The overall English needs to be improved. Please seek guidance from a native English speaker if possible ("the" "a", commas, plural form, and others could be corrected).
2) The authors could insert more numerical data into the Abstract for enhancement of the manuscript.
3) Please better explain the novelty of your work.
4) The introduction is well-written but not sufficient. More papers need to be cited and discussed. Please add some information about the other applications of graphene materials. Please cite:
(1) Nanomaterials. 2019; 9 (12): 1758. DOI: 10.3390/nano9121758
(2) Int. J. Mol. Sci. 2023, 24(4), 3576; https://doi.org/10.3390/ijms24043576
(3) Nanomaterials 2021, 11(8), 2080; https://doi.org/10.3390/nano11082080
5) Figure 3. SEM images please add more SEM with different magnifications.
6) Figure 5d please correct this image (description) for better quality.
7) Could the authors include the standard deviation of the used analysis?
8) Why authors do not use an XPS for potential group characterization and elements?
9) Why author choose these materials for the study, not other carbons? Please explain.
10) What the authors said about the quality aspect of materials?
11) Authors are suggested to describe some potential applications, where we can use this knowledge. Please write 1-2 sentences in conclusions to enhancement.
12) Please see the structure of the manuscript in Nanomaterials.
The overall English needs to be improved. Please seek guidance from a native English speaker if possible ("the" "a", commas, plural form, and others could be corrected).
Round 2
Reviewer 3 Report (New Reviewer)
The authors have addressed all comments and the manuscript can be published as is.
This manuscript is a resubmission of an earlier submission. The following is a list of the peer review reports and author responses from that submission.
Round 1
Reviewer 1 Report
The submitted paper presents an interesting study. Yet the general results obtained are relatively incremental compared to those of the studies cited in the manuscript. The main achievements of the study are using PECVD, graphene nanowalls (GNWs) instead of planar graphene, and the insertion of a high-permittivity HfO2 interface layer between GNWs and doped Si.
My reading of the paper arose several questions and doubts, and I would like the authors to answer and resolve them. Let me note them going through the paper's sections.
Introduction. Here I've acquired the impression that authors in some places do not present physically correct formulations. For example, what does it mean in lines 26, and 27: "... exceptional electrical and optical qualities like ..."? In this manner, physicists do not write; this is also not correct English. Even more physically illiterate is the phrase in line 28: "...giant intrinsic mobility, leading to zero effective mass ..." How it can be? The relationship between mobility and mass is the converse of what the authors claim. Line 83, what does "... the energy band of the device..." mean? There is no such a notion, it is rather the band diagram. Last but not least, the authors do not clearly explain what is GNWs and why their use is more favorable as compared to that of graphene sheet (exfoliated or CVD fabricated).
Experimental. What does the abbreviation RCA in line 88 mean? Figs.1 and 2 are too schematic, it would be better to expose real apparatus and intermediate fabrication products. But using the rich instrumentary declared is laudable.
Results and Discussion. Figs.3 (a) - (d) show that GNWs are structurally inhomogeneous. The inhomogeneity impact on light transmission is inquired. But also, they hardly have mobility as high as that of graphene sheets which is more important for electronic devices made of GNWs. What about this issue? Fig. 4 (a) - a pronounced singularity of the I-V curve at zero bias is seen. What physics is behind it? Fig. 5 - the substantiation of the band diagram is by far insufficient. More extended discussion is welcome.
Reviewer 2 Report
This work, from Y. Shen and co-workers titled “High Performance Graphene Nanowalls/Si Self-powered Photodetectors with HfO2 as an Interfacial Layer” describe the realization of photodetector using graphene nanowalls (GNWs) silicon heterojunction, with the introduction of a thin hig-dielectric material at the interface.
The photodetector characterized in this work are made by means of ALD-growth hafnium dioxide and PECVD-growth GNWs. The device geometry is defined with optical lithography, metal sputtering and oxygen plasma. Preliminary characterization of the GNWs have been made via SEM, Raman and optical transmittance. Finally, the photodetector properties have been proved in DC configuration, with optical excitation in the visible range (532 nm).
However, there are many concerns in this work:
-
The introduction can be improved, especially it is not clear the paragraph regarding the growth of graphene, since most of the reference are about GNWs.
-
Some references are missing in the introduction, especially when some relevant parameter of the material (i.e. is reported the optical absorption of graphene, without citing Nair2008[10.1126/science.1156965] or other works). Also, in several point the authors refer to review works instead of original papers.
-
The performance parameters reported are not so outstanding, moreover the authors don’t compare with Yang2021 [(10.1039/c7nr00573c], moreover the responsivity reported in table 3 for X. Liu 2019 [37] it wrong.
-
A more detailed characterization of the HfO2 should be performed: the thickness of the layer should be measure independently (via AFM or profilometer) and compared with the number of cycles. Also, a morphologic characterization of the material (as adding a SEM of the bare HfO2 in fig 3) should be performed. Finally, some electrical characterization of the material, as the dielectric constant and breakdown voltage should be added at least in supporting.
-
The growth of GNW should be described more in detail, a comparison with the literature is missing: it is not clear if the reported growth represents an improvement to the state of the art or not.
-
Should be added more details in the experimental section regarding the device fabrication: what kind of resist has been used? What is the power of the UV exposure? Moreover an image of the final device, with the characteristic size should be added in supporting information.
-
The discussions on the Raman should be carefully reviewed: are missing the references, the reported value of the G peak (1690 cm-1) is quite far from standard 1580 cm-1, the claim of absence of D’ peak should be better discuss, since can be present as a shoulder of the G peak and the difference between 60 seconds and 90 seconds growth shown in fig 3e should be better discuss, maybe adding an intermediate growth time.
-
The figures should be improved, both in the arrangement and in the labels, as an example the figure 4a is not clear and the labels seems wrong.
-
The description of the electrical characterization can be improved: it is not clear what growth-time of GNWs has been used. Moreover, the time response characterization reported in fig 4b seems limited by the acquisition time resolution.
-
Are missing the references for the formulas to obtain the performance parameter, and the quantum efficiency values should be compared with the state of the art.
The work regards a relevant topic in the field of graphene/based materials and photodetector device, however is missing any factor of novelty to justify the publication in this form. I would suggest the authors to perform some more detailed characterization before re-submitting the paper.
Reviewer 3 Report
The authors have demonstrated the benefit of the HfO2 interfacial layer in the photodetector by several experimental methods. They have also shown the advantage of the remote plasma method for growing the graphene material. Results include a very good improvement in quantum efficiency.
The paper will be good for publication after the following minor revisions.
1. In Fig. 4(a) there is some confusion about which curve represents which condition. The legend may need to be corrected. Also, in the PDF version, this reviewer is not able to see the difference between the markers on the two red curves, and the difference between the markers on the two black curves.
2. On lines 271 and 272 the paper needs a brief explanation (one or two sentences) as to why the holes have a higher probability than electrons for tunneling through the barrier.
3. On line 284, in the units for the Richardson constant, k should be capital K for Kelvin.
Reviewer 4 Report
This manuscript presented a photodetector device based on graphene/Si heterojunction with an inter-layer of HfO2. The authors used chemical vapor deposition with lower power to avoid damage by processing the interfacial properties; therefore reported high performances attributed to the interested HfO2 layer. The authors’ device offered self-bias operation, which is crucial for developing sustainable optoelectronic applications. However, severe changes and a few other results are needed. My comments to improve this work are summarized below.
1. Why authors use n-Si as a photoactive material should be given to the reader? Why not p-Si or other photoactive materials?
2. Is there any specific reason for the metal contacts of Cr/Au on the front side? Readers may need this information and help them to refer to this study.
3. The value for the processing power of 300 W should be provided in the power density.
4. The authors mentioned the excellent electrical conductivity of GNW. Please add electrical results, such as conductivity and sheet resistance, to Figure 3.
5. Due to the photodetection application, authors should provide information on the illumination wavelength. Readers may wonder what the wavelength-dependent performance of the authors’ device is. Please add the intensity of light illuminations.
6. Why the response time of the Si device with an HfO2 layer is of the order of ms should be discussed and given to the readers.
7. Optical spectra of the HfO2 layer and its band-gap value should be given to support the claim.
8. Of curiosity, can the author describe what happens if they prepare a device with back contact to the Si wafer?
9. For the topic relative to this work, the reports of the application of Si Schottky photodetectors in Appl. Phys. Lett. 108 (2016) 141904, Physica B 537 (2018) 228–235; Si heterojunction based on 2D semiconductors in ACS Appl. Mater. Interfaces 10(4) (2018) 3964–3974, Nanoscale 9 (2017) 15804-5812; and other Si photodetectors in ACS Nano 11 (2017) 10955–10963, ACS Appl. Mater. Interfaces 14 (2022) 32341–32349, Adv. Mater. Technol. (2022) 2200863; are valuable to refer.
Authors may improve their English for readability.
Thank you.